# The Basin Stability of Bi-Stable Friction-Excited Oscillators

**Merten Stender** [1,*] **, Norbert Hoffmann** [1,2] **and Antonio Papangelo** [1,3]

[1]  Hamburg University of Technology, 21073 Hamburg, Germany; norbert.hoffmann@tuhh.de (N.H.); antonio.papangelo@poliba.it (A.P.)
[2]  Imperial College London, London SW7 1AL, UK
[3]  Politecnico di Bari, 70125 Bari, Italy
[*]  Correspondence: m.stender@tuhh.de

**Abstract:** Stability considerations play a central role in structural dynamics to determine states that are robust against perturbations during the operation. Linear stability concepts, such as the complex eigenvalue analysis, constitute the core of analysis approaches in engineering reality. However, most stability concepts are limited to local perturbations, i.e., they can only measure a state's stability against small perturbations. Recently, the concept of *basin stability* was proposed as a global stability concept for multi-stable systems. As multi-stability is a well-known property of a range of nonlinear dynamical systems, this work studies the basin stability of bi-stable mechanical oscillators that are affected and self-excited by dry friction. The results indicate how the basin stability complements the classical binary stability concepts for quantifying how stable a state is given a set of permissible perturbations.

**Keywords:** nonlinear dynamics; basin of attraction; self-excitation; bi-stability; multi-stability

## 1. Introduction

The dynamics of systems affected by friction are most often studied in the context of friction-excited vibrations (FIV). Prominent examples for FIV in mechanical structures and machines range from brake systems [1–4], clutches [5], drill strings [6] to artificial hip joints [7] and others. FIV often arise through positive energy feedback from a friction interface with the structure, i.e., through self-excitation [8–10]. Sub-critical Hopf bifurcations [11,12] and isolated solution branches [13–15] are a common observation in those systems, such that bi- and multi-stable systems have been reported numerously [14,16,17]. The computation of those nonlinear responses (periodic, quasi-period orbits, chaotic trajectories) is a well-established field of research [18–21], mostly resulting in the identification of complicated bifurcation diagrams [11,13,22–24]. The stability of the solutions is usually assessed by local Lyapunov-type stability metrics [25,26]. Hence, the stability statement is often a binary one that measures the state's robustness against *small* perturbations. However, the actual size of *permissible perturbations*, i.e., those for which the trajectory would still return back to the state, is not given. In a multi-stable system configuration, the long term steady-state behavior thus depends on the choice of initial conditions or the size of instantaneous perturbations. Once the system enters another basin of attraction, severe jumping phenomena may occur. Typically, such jumps are related to a change from a stable steady sliding state to high-intensity periodic vibrations or stick-slip cycles [27–29], or from one periodic solution to another periodic solution [11].

This work investigates a rather novel technique denoted as *basin stability* to estimate the size of the system's basins of attraction in a subset of the state space. The basins' size estimation can then be considered a global stability metric, i.e., indicating how likely the system is to end up on one of the

co-existing stable steady-states. Therefore, those probabilities add new insights to the rather binary stability statements derived from local perturbation-based approaches. We study a friction oscillator excited by a falling friction slope and a second oscillator excited through binary flutter instability. Our results indicate that the basin stability analysis is a robust and easily applicable model-agnostic technique. It can reveal the *actual* picture of the long-term behavior for a given set of perturbations, thus augmenting classical bifurcation and stability studies. Using the basin stability analysis, some solutions can even be ruled out if one can guarantee strict control over the instantaneous perturbations to system trajectories or operating conditions.

## 2. The Concept of Basin Stability

We study nonlinear dynamic systems

$$\dot{\mathbf{x}} = \mathbf{f}(\mathbf{x}, t), \quad \mathbf{x} \in \mathbb{R}^N \tag{1}$$

with the states $\mathbf{x}(t)$ in the $N$-dimensional state space. The long-term asymptotic behavior is denoted as *attractor* $\mathcal{A}$ [30] throughout this work. Typically, the Lyapunov spectrum $\Lambda = [\lambda_1, \dots, \lambda_N]$ is assessed to characterize the linear stability of a state $\mathbf{x}$ against small perturbations. For fixed points, the Lyapunov exponents are equivalent to the system's eigenvalues derived from the complex eigenvalue analysis (CEA). The real parts of the eigenvalues indicate linear stability to a *small* perturbation about the fixed point. The sizes of the real parts indicate the strength of attraction ($\lambda < 0$) or rejection ($\lambda > 0$) for stable or unstable directions in state space, respectively. However, the eigenvalues do not encode a piece of information about the permissible size of perturbations that are still attracted by the fixed point. While this is not an issue for systems that feature only a single stable solution, the situation is different for systems featuring multiple stable solutions. For these systems, local stability concepts may have only a limited validity: a non-small perturbation of a state can result in a jump to another attractor. Hence, global stability concepts are required to assess the size of permissible perturbations, i.e., to characterize the basins of attraction for all solutions. The basin of attraction

$$\mathcal{B}(\mathcal{A}) = \left\{ \mathbf{x}_0 \in \mathbb{R}^N \mid \lim_{t \to \infty} \mathbf{x}(t) = \mathcal{A}, \quad \mathbf{x}_0 = \mathbf{x}(t=0) \right\} \tag{2}$$

denotes the subset of states that converge to the same attracting set $\mathcal{A}$. The basin boundaries are related to unstable solutions of the system which represent separatrices of the basins in state space. Depending on the size and shape of its basin, an attractor can be more or less robust against non-small perturbations. There are multiple ways to compute the basins of attraction, e.g., through Lyapunov functions [31]. These methods are known for some canonical, low-dimensional, and well-studied systems. However, they are not readily available, or straight-forward to compute, for any generic and high-dimensional nonlinear dynamical system, such as frictional oscillators which are studied in this work.

The basin stability proposed by Menck et al. [32,33] is a global stability concept for complex systems that aims at measuring stability against non-small perturbations by a volume-based probabilistic approach. Conceptually, the basin stability measures the volumetric share of all basins of attraction in a hypervolume of the state space. For a computationally feasible solution, a distribution $\rho(\mathbf{x})$ of perturbations is drawn from a reference subset $\mathcal{Q} \subset \mathbb{R}^N$ of the state space, representing a set of states to which the system may be pushed to through non-small perturbations with $\int_{\mathcal{Q}} \rho(\mathbf{x}) \, d\mathbf{x} = 1$. For each perturbation, the steady-state behavior of the dynamical system is obtained through time-marching integrations. Then, the fraction of perturbed states that converged to the specific attractor $\mathcal{A}$ denotes an estimate for the basin stability $\mathcal{S}_{\mathcal{B}}(\mathcal{A})$, i.e., [32,34]

$$\mathcal{S}_{\mathcal{B}}(\mathcal{A}) = \int \kappa_{\mathcal{B}}(\mathcal{A})(\mathbf{x}) \rho(\mathbf{x}) \, d\mathbf{x}, \quad \kappa_{\mathcal{B}}(\mathcal{A})(\mathbf{x}) = \begin{cases} 1, & \text{if } \mathbf{x} \in \mathcal{B}(\mathcal{A}) \\ 0, & \text{otherwise} \end{cases}. \tag{3}$$

Here, $\kappa_{\mathcal{B}}$ denotes an indicator function that classifies a steady state solution $\mathbf{x}(t)$ to belong to the attractor $\mathcal{A}$. Therefore, $\mathcal{S}_{\mathcal{B}}(\mathcal{A})$ is an estimate for the volume share of the basin of attraction $\mathcal{B}_{\mathcal{A}}$ given the reference subset $\mathcal{Q} \subset \mathbb{R}^N$ sampled by $\rho(\mathbf{x})$ [32,35]. Naturally, for a $k$-multi-stable system, the basin stability values of all $k$ attractors add up to unity $\sum_i^k \mathcal{S}_{\mathcal{B}}(\mathcal{A}_i) = 1$. The size, i.e., the number of states, of the dynamical system to be studied by the basin stability is only limited by computational power for the Monte Carlo simulations. As the basin stability computation can be considered a repeated Bernoulli experiment [32], the standard error of the basin stability estimate is

$$e = \frac{\sqrt{\mathcal{S}_{\mathcal{B}}(1 - \mathcal{S}_{\mathcal{B}})}}{\sqrt{n}} \tag{4}$$

which can be used to find a subset $\mathcal{Q}$ that ensures a low standard error. Recently, systems with fractal, riddled, and intermingled basin boundaries were studied [33] indicating the robustness of the basin stability concept. All basin stability computations in this work were obtained from the open-source package `bSTAB` [36] available at https://github.com/TUHH-DYN/bSTAB/tree/v1.0.

Figure 1 displays a schematic for illustrating the practical computation of basin stability values. A nonlinear dynamical system with two states $\mathbf{x} = [x_1, x_2]$ is studied (In fact, the system is the single-degree-of-freedom frictional oscillator to be discussed in Section 3). The system exhibits three solutions: A stable equilibrium position ($\mathbf{x}_{EP}$), an unstable periodic orbit ($\mathbf{x}_{UPO}$), and a stable limit cycle ($\mathbf{x}_{LC}$). The distribution of perturbations $\rho(\mathbf{x})$ is chosen such that all solutions are contained in $\mathcal{Q}$ and $n = 100$ samples are drawn uniformly at random. The trajectories starting from $n_{EP} = 37$ states in the basin $\mathcal{B}_{EP}$ converge towards the equilibrium position, while $n_{LC} = 63$ states are located in the basin $\mathcal{B}_{LC}$ and thus converge to the stable limit cycle. As a result, the basin stability estimates are $\mathcal{S}_{\mathcal{B}}(EP) = 0.37$ and $\mathcal{S}_{\mathcal{B}}(LC) = 0.63$, respectively. Because the separatrix, which is the unstable periodic orbit, is explicitly known for the system, the basin volumes can be determined analytically. The exact volumetric fractions of $\mathcal{B}_{EP}$ and $\mathcal{B}_{LC}$ in $\mathcal{Q}$ are 0.3275 and 0.6725, respectively. Therefore, the basin stability computed from $n = 100$ samples is a good approximation for the system at hand (Appendix A.2 indicates that $n \approx 300$ samples are required for a very close approximation of the analytical results).

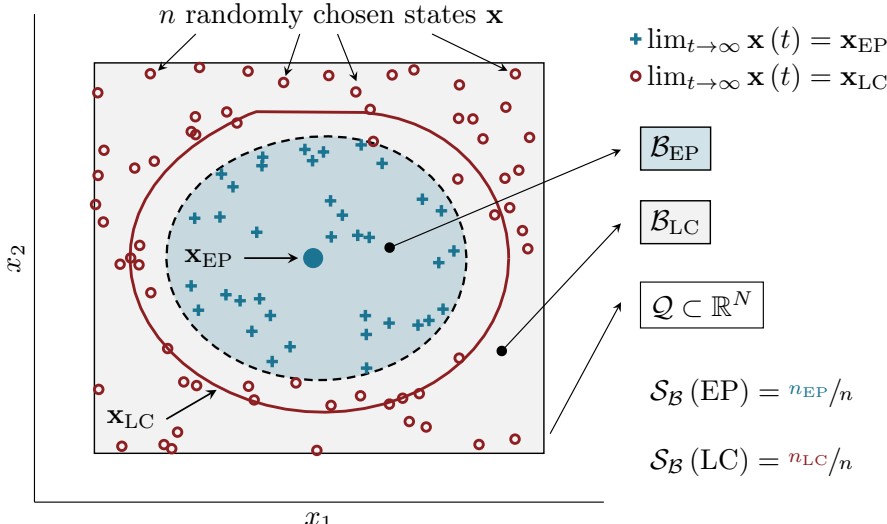

**Figure 1.** Schematic of the basin stability calculation. In the two-dimensional state space, two stable attractors EP (equilibrium position) and LC (limit cycle) co-exist. The respective basins of attraction $\mathcal{B}_{EP}$ and $\mathcal{B}_{LC}$ are separated by an unstable periodic orbit (indicated by the dashed line). The steady-state behaviors of $n = 100$ randomly sampled states are used to estimate the volume shares of the basins of attraction in the subset $\mathcal{Q}$. The resulting basin stability estimates are $\mathcal{S}_{\mathcal{B}}(EP) = 0.37$, $\mathcal{S}_{\mathcal{B}}(LC) = 0.63$ for this example.

## 3. Bi-Stable Oscillator with Falling Friction Slope

As a first system, we study the dynamics and the stability of a single-degree-of-freedom oscillator $m\ddot{x} + c\dot{x} + kx = F$, see Figure 2a, with velocity-dependent friction as proposed by Papangelo et al. [12]. Specifically, the friction characteristic $\mu(v_{\mathrm{rel}})$ is a velocity-dependent weakening function

$$
\begin{aligned}
v_{\mathrm{rel}} \neq 0 : \quad & F = -N\mu(v_{\mathrm{rel}})\,\mathrm{sign}(v_{\mathrm{rel}}), \quad v_{\mathrm{rel}} = \dot{x} - v_{\mathrm{d}} \\
v_{\mathrm{rel}} = 0 : \quad & |F| < \mu_{\mathrm{st}} N \\
& \mu(v_{\mathrm{rel}}) = \mu_{\mathrm{d}} + (\mu_{\mathrm{st}} - \mu_{\mathrm{d}})\exp\left(-\frac{|v_{\mathrm{rel}}|}{v_0}\right)
\end{aligned}
\tag{5}
$$

featuring the static friction coefficient $\mu(0) = \mu_{\mathrm{st}}$, the dynamic friction coefficient $\mu(v_{\mathrm{rel}} \to +\infty) = \mu_{\mathrm{d}}$, the reference velocity $v_0$ and the contact normal load $N$. The non-dimensional form of the equations of motion is obtained through normalization $(\tilde{\cdot})$ of the quantities accordingly to the work of Papangelo et al. [12]. The velocity-dependence introduces a dynamic instability that gives rise to friction-induced vibrations (FIV) for $0 \leq \tilde{v}_{\mathrm{d}} \leq 1.84$. Moreover, the friction nonlinearity enables the system to exhibit a bi-stable behavior, such that a stable steady sliding state and a stable stick-slip cycle co-exist for a range of belt velocities $1.11 \leq \tilde{v}_{\mathrm{d}} \leq 1.84$, see Figure 2b. At $\tilde{v}_{\mathrm{d}} = 1.15$, the steady sliding state loses stability at a subcritical Hopf bifurcation point. In the bi-stability regime, and depending on the initial condition or instantaneous perturbations, the system will either end up in the low-energy steady sliding state, or on the high-intensity stick-slip cycle. Both solutions are locally stable and attractive, i.e., robust against small perturbations.

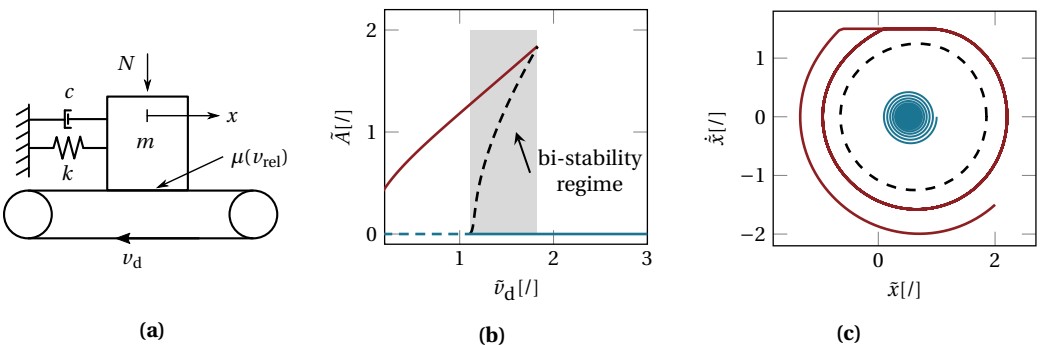

**(a)**    **(b)**    **(c)**

**Figure 2.** (**a**) single-degree-of-freedom frictional oscillator, (**b**) bifurcation diagram for the non-dimensional belt velocity $\tilde{v}_{\mathrm{d}}$, and (**c**) phase plane for $\tilde{v}_{\mathrm{d}} = 1.5$. Stable (unstable) solutions are indicated by solid (dashed) lines. The stable steady sliding state (blue spiral trajectory) co-exists with the unstable periodic orbit (black dashed line) and the stable stick-slip limit cycle (red trajectory). The non-dimensional system $(\tilde{\cdot})$ is evaluated for $\mu_{\mathrm{d}} = 0.5$, $\mu_{\mathrm{st}} = 1.0$, $\xi = 0.005$, $N = 1.0$ and $\tilde{v}_0 = 0.5$.

For this minimal system, the basin boundaries are directly accessible through the unstable periodic orbit (UPO). However, if this knowledge was not available, the probability of arriving on one of the two steady states would be unknown. Figure 1 displays a sampling with $n = 100$ points uniform at random from $\mathcal{Q}(x, \dot{x}) : [-3, 3] \times [-2, 2]$ at $\tilde{v}_{\mathrm{d}} = 1.5$, and the resulting basin stability values $\mathcal{S}_{\mathcal{B}}(\mathrm{FP}) = 0.37$ and $\mathcal{S}_{\mathcal{B}}(\mathrm{LC}) = 0.63$. Hence, for this $\rho(\mathbf{x})$, it is more likely to arrive on the high-amplitude limit cycle solution than on the steady sliding fixed point.

To complement the bifurcation diagram and the complex eigenvalue analysis, the basin stability of the fixed point and limit cycle solution is derived along the normalized belt velocity parameter. In particular, at each velocity value $n = 1000$ initial conditions are drawn from a uniform random distribution in $\mathcal{Q}(x, \dot{x}) : [0.5, 2.5] \times [-2, 0]$, i.e., positive initial displacement and negative initial velocity. Figure 3 depicts the eigenvalue's real part and the basin stability. As $\tilde{v}_{\mathrm{d}}$ decreases, the real part grows until it crosses into the positive plane at $\tilde{v}_{\mathrm{d}} = 1.15$. This rather smooth behavior nicely indicates the transition into linear instability of the fixed point solution. However, the eigenvalues at

the exemplary points $\tilde{v}_\mathrm{d} = 1.3$ and $\tilde{v}_\mathrm{d} = 1.7$ would not allow a statement about the system's probability to converge to this solution instead of converging to the periodic orbit. Additionally, the eigenvalue obviously does not indicate the existence of the competing stable periodic solution in this parameter range. At this point the basin stability analysis comes into play: Below the Hopf bifurcation point, all trajectories converge to the periodic orbit, hence $\mathcal{S}_\mathcal{B}\,(\mathrm{LC}) = 1.0$, and above the bi-stability regime all trajectories converge to the globally stable fixed point, i.e., $\mathcal{S}_\mathcal{B}\,(\mathrm{FP}) = 1.0$ for $\tilde{v}_\mathrm{d} > 1.84$. For the chosen subset $\mathcal{Q}$, the periodic orbit is the dominating behavior in the lower parameter range of the bi-stable regime. For increasing relative velocity the probabilities, i.e., the basin stability values, are more balanced for arriving either on the LC or the FP. For $\tilde{v}_\mathrm{d} > 1.6$ the fixed point is the more probable solution to arrive at. Therefore, the basin stability values add an important insight and complement the binary stability statements given by the eigenvalues. Using the basin stability, it is now possible to state *how* stable a solution is against arbitrary and possibly *non-small* perturbations. For more realistic systems, this statement may be of even larger value than the binary stability statement given by local metrics.

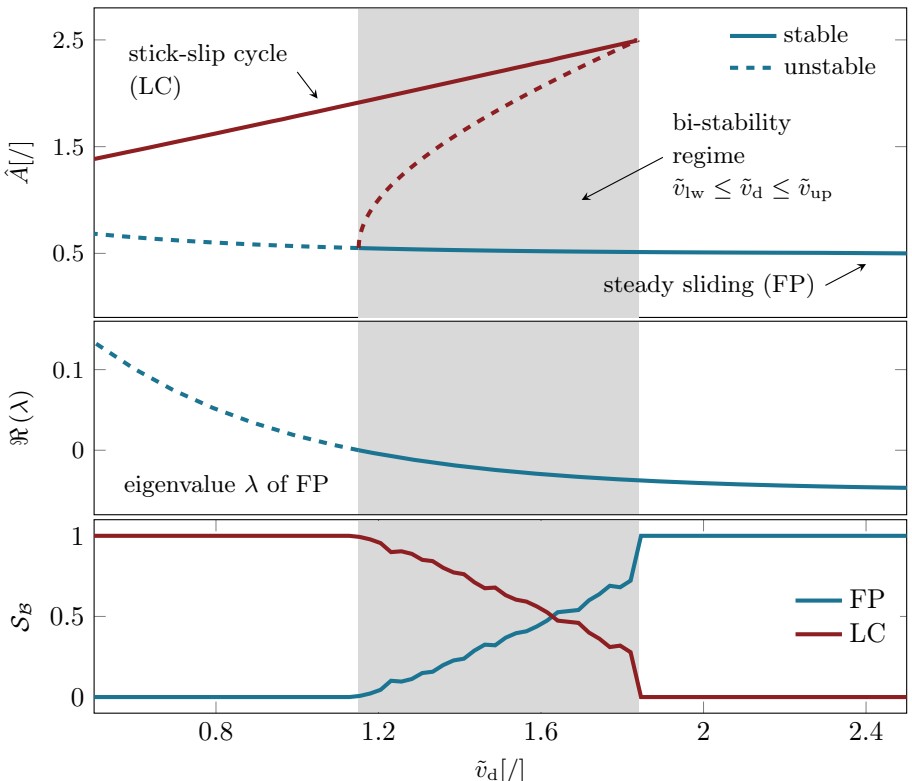

**Figure 3.** Bifurcation diagram (**top**), real eigenvalue (**middle**) and basin stability (**bottom**) of the single-DOF friction oscillator along the relative sliding velocity.

## 4. Bi-Stable Oscillator with Mode-Coupling

As a second system, we study a frictional oscillator [8,11], which (in contrast to the first system) experiences FIV through a mode-coupling instability. The system features a main oscillating mass that is in dry Coulomb-type frictional contact with a conveyer belt. A second mass is connected to the main mass through a nonlinear joint element in diagonal direction, thereby geometrically coupling the vertical and horizontal movement of the main mass. The relative sliding velocity is assumed to always be positive, such that no stick-slip cycles can arise. For the nonlinear joint element, a cubic stiffness nonlinearity $k_\mathrm{nl}$ is chosen [11]. The equations of motion and parameter values are given in Appendix B and the model is displayed in Figure 4.

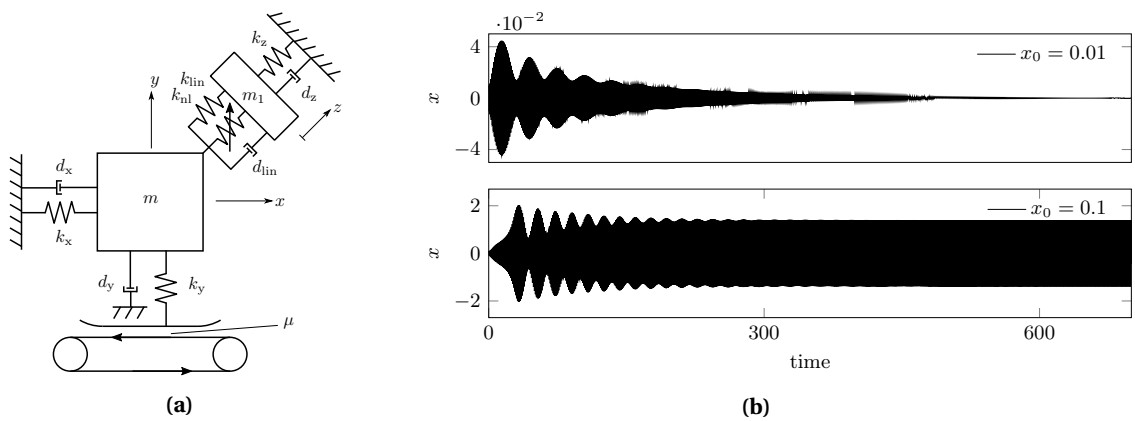

**Figure 4.** (**a**) Frictional oscillator with nonlinear joint and mode-coupling instability [11]. (**b**) Trajectories obtained in the reference configuration (see Appendix B) for two different initial conditions of the horizontal displacement $x$ (all other states were kept at 0).

Previous studies have revealed the complicated bifurcation behavior of this system, including super- and sub-critical Hopf bifurcations as well as isolated solution branches [11,13,14]. In this study, a variation of the horizontal stiffness $k_x$ is performed. A sub-critical Hopf bifurcation point is found at $k_x = 32.3$, see Figure 5a. Below, a stable limit cycle and the unstable fixed point exist. Above this value there is a bi-stable range up to $k_x = 33.0$ with a co-existing stable limit cycle and the stable fixed point. The eigenvalues' real parts in Figure 5b exhibit the classical forking behavior that is related to the mode-coupling instability mechanism in this system. At the point of instability, one eigenvalue crosses into the positive plane. The basin stability $\mathcal{S}_B$ of both stable solutions is computed for $n = 500$ random initial conditions drawn from $\mathcal{Q}(x, \dot{x}) : [0, 0.5] \times [0, 0.25]$ (all other initial conditions are fixed to 0). Figure 5c depicts the basin stability as a function of the horizontal stiffness. In the bi-stability range $32.3 \leq k_x \leq 33.0$ the basin stability values indicate that the limit cycle solution is the dominating one for lower stiffness values. For larger stiffness values the fixed point solution is the most probable for our choice of $\mathcal{Q}$. Hence, within this rather short bi-stability range, a minor variation of the horizontal stiffness value would crucially affect the probability of arriving either on the low-energy steady-sliding state, or on the high-energy limit cycle, which may cause increased wear, audible vibrations and other effects in realistic systems. Such kind of statement about the global stability regarding non-small perturbations would not have been easily accessible through the bifurcation diagram or the local stability analysis.

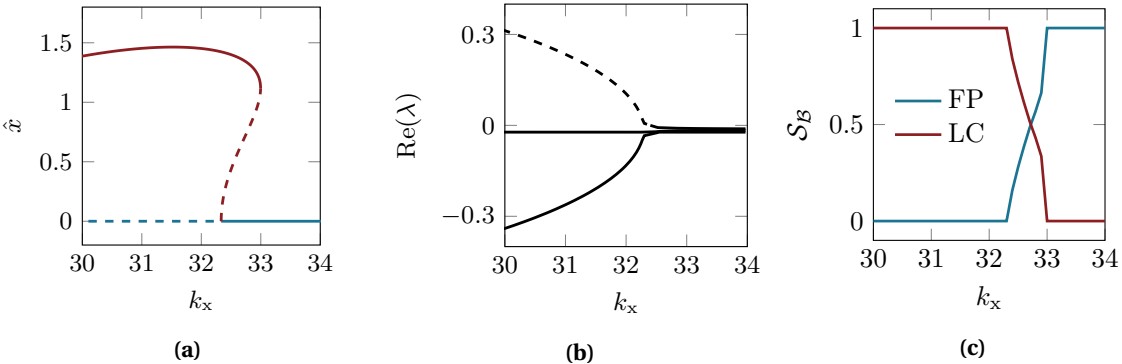

**Figure 5.** (**a**) bifurcation diagram for the horizontal stiffness parameter, (**b**) eigenvalues' real parts and (**c**) basin stability of the fixed point and limit cycle solution. $\hat{x}$ denotes the maximum amplitude of $x(t)$ along one vibration period. Solid and dashed lines indicate stable and unstable solutions, respectively.

These results are clearly related to the shape of the unstable periodic orbit, i.e., the separatrix of both basins of attraction. While the qualitative basin stability values for a variation in the initial

conditions for $x$ would have been readable from the bifurcation diagram in Figure 5a, this task quickly becomes complex once more degrees-of-freedom (DOFs) are considered. For example, let us consider a reference subset $\mathcal{Q}$ that captures certain variations for multiple DOFs, instead of variations for a single DOF as shown before. Figure 6 displays the basin stability values for three different choices of $\mathcal{Q}$, i.e., different variations of the range of possible initial conditions:

$$
\begin{aligned}
&\mathcal{Q}_1\,(x,y) : [0, 0.25] \times [0, 0.5]\\
&\mathcal{Q}_2\,(x,y,\dot{x},\dot{y}) : [0, 0.25] \times [0, 0.25] \times [-0.1, 0.1] \times [-0.2, 0.2]\\
&\mathcal{Q}_3\,(x,y,z,\dot{x},\dot{y},\dot{z}) : [0, 0.25] \times [0, 0.25] \times [0, 0.25] \times [-0.1, 0.1] \times [-0.1, 0.1] \times [-0.1, 0.1]\,.
\end{aligned}
\tag{6}
$$

In the first case, some initial variations in the horizontal position and large variations in the vertical displacement of the main mass are allowed. In the second case, variations in the initial velocities are studied, and in the third case also variations in the secondary mass' initial conditions are considered. Such scenarios would quickly become impractical for studying permissible perturbations, i.e., the global stability of each solution, using bifurcation diagrams and subdividing the state space by the unstable solutions. The concept of basin stability automates this process through the Monte Carlo sampling, allowing for a easy-scaling and consistent estimation of the relevant basin volumes.

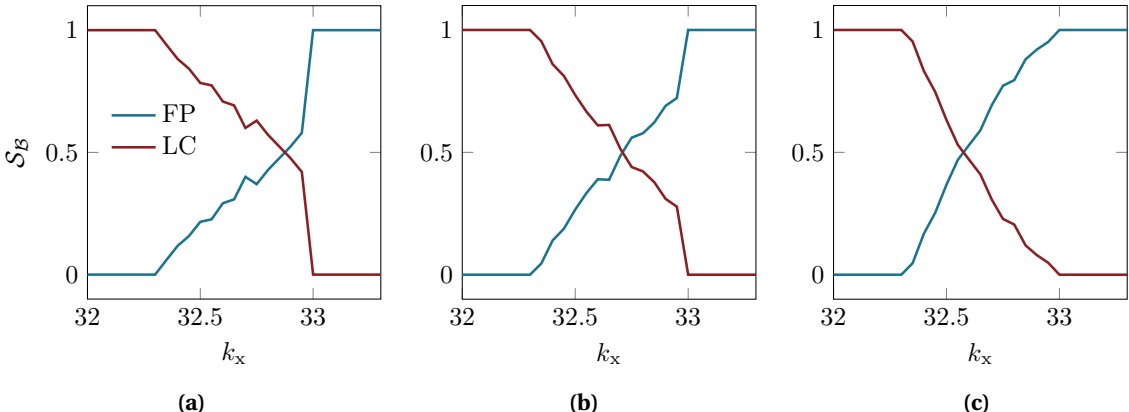

**Figure 6.** Basin stability values in the bi-stability range for the reference sets of initial conditions $\mathcal{Q}_1$ (**a**), $\mathcal{Q}_2$ (**b**), and $\mathcal{Q}_3$ (**c**) defined in Equation (6).

In fact, even though the three reference sets are very different in their value ranges, the resulting basin stability analysis displayed in Figure 6 does not change qualitatively. The *turning point*, i.e., the point after which the FP solution dominates over the LC solution for increasing values of $k_x$, changes only slightly: For $\mathcal{Q}_1$ this point is found at $k_x = 32.9$, while it is $k_x = 32.7$ and $k_x = 32.55$ for $\mathcal{Q}_2$ and $\mathcal{Q}_3$, respectively. Hence, the basin stability is not very sensitive to the choice of $\mathcal{Q}$ for this system. In a situation in which the overall qualitative behavior of the basin stability values may have seem obvious, the quantitative evaluation would have become difficult to obtain from the bifurcation diagrams. Especially for higher-dimensional systems and specific subset choices the basin stability analysis represents a highly robust approach to estimate the probability of arriving on either of the competing solutions, which we will illustrate in the next section.

## 5. Bi-Stable Oscillator with Isolated Periodic Solution

The third dynamical system studied in this work is a weakly damped variant of the system proposed in the previous section and sketched in Figure 4. Here, the damping parameters are reduced by a factor of 10 to $d_x = d_y = d_z = d_{\text{lin}} = 0.002$. This system configuration has already been studied in [13,14] where the authors found an isolated solution branch resulting from the damping variation. Figure 7a displays the bifurcation diagram for the horizontal stiffness $k_x$. The fixed point solution loses stability through a sub-critical Hopf bifurcation at $k_x = 32.24$ to a limit cycle solution, hereafter denotes

as LC1. Interestingly, a second stable limit cycle solution is born for $k_x < 29.9$, which is found to be an isolated branch [14], hereafter denoted as LC2. That is, this solution is not connected to any other solution path. As a result, the system may jump from the fixed point solution to the first limit cycle for $32.24 \leq k_x \leq 33.0$, and then from the limit cycle to the isolated branch for $k_x < 29.9$. Hence, within a rather narrow parameter range, two jumping phenomena between different solutions may occur. It is, therefore, of great interest to investigate the probability of arriving on either of those solutions for some prescribed set of initial conditions.

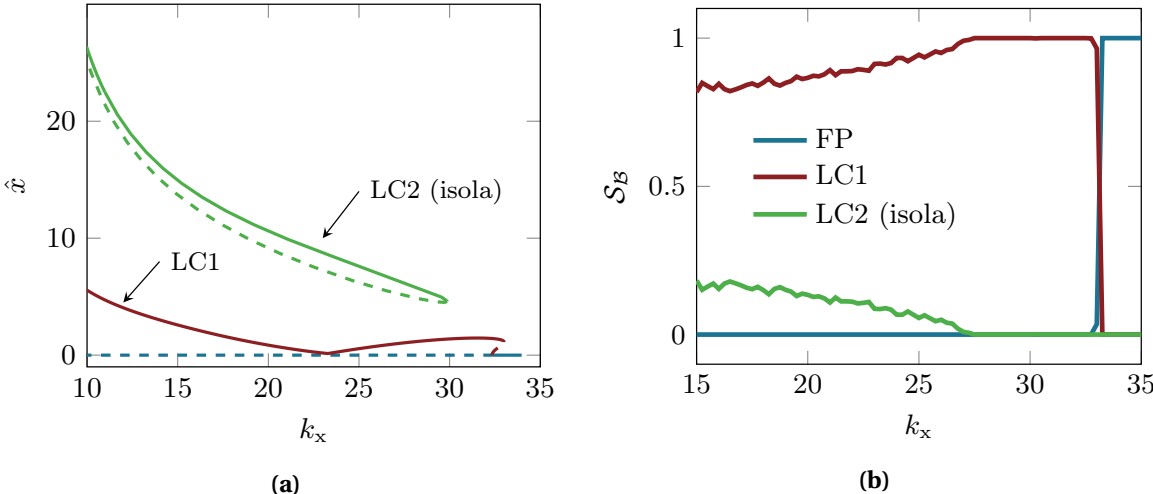

**Figure 7.** Bifurcation diagram for the weakly damped friction oscillator exhibiting an isolated solution branch (**a**) and basin stability values (**b**) for all three stable solutions along the horizontal stiffness $k_x$. Initial conditions for each solution are given in Appendix C.

Figure 7b displays the basin stability values for both periodic orbits and the fixed point solution. For the reference subset, we arbitrarily chose $\mathcal{Q}(x, y, z, \dot{x}, \dot{y}, \dot{z}) : [0, 10] \times [0, 10] \times [0, 10] \times [-2, 2] \times [-2, 2] \times [-2, 2]$ using $n = 1000$ sampling points. For for the bi-stability range featuring the two periodic solutions LC1 and LC2 ($k_x < 29.9$) the basin stability analysis reveals that LC1 is the by far most probable solution. A maximum of 21% of the trajectories converge to the isolated solution branch, while the remaining trajectories converge to the first periodic orbit. Particularly interesting is the parameter regime $27.4 \leq k_x \leq 29.9$. Here, the basin stability indicates that LC1 is globally stable, even though the stable isola still co-exists. However, due to the choice of $\mathcal{Q}$, no initial conditions were drawn for the basin related to LC2. Hence, if the range of initial conditions and perturbations can be quantified or limited for some specific system, the basin stability analysis can also help to rule out jumping phenomena between co-existing solutions.

Another interesting observation is the following: the basin stability values in this specific setting do not follow the qualitative trend of the respective amplitudes reported in Figure 7a. $\mathcal{S}_B$ (LC1) keeps increasing along the stiffness parameter, while the corresponding amplitude of the horizontal vibration amplitude shows a different behavior. Theoretically, it is clear that the vibration amplitudes do not relate to the size of the basins of attraction. However, on the first sight classical bifurcation diagrams may suggest that one solution is *more attractive* if it has a larger vibration amplitude. At this point, the basin stability represents a technique to quantify the attractiveness in a highly consistent manner.

Lastly, we discuss our previous thought on the benefits of having a robust methodical approach to estimating the basin volumes through Monte Carlo sampling irrespective of the dynamical system at hand (so-called *model-agnostic* techniques). Especially for such low-dimensional systems as shown before, one might raise the issue of using computation-heavy sampling methods, even though the basins of attraction are readily available once the bifurcation diagram is known. Figure 8 displays the state space of each DOF at $k_x = 27$, hence in a configuration where the two periodic orbits co-exists. It becomes clear that even for this 3 DOF oscillator (6 states), the analytical calculation of the basin

volumes can quickly become a challenge. There is no straight-forward way to computing the volumes in the six-dimensional space from the intertwined basins separated by the unstable orbits, especially looking at the $z$ coordinate. Therefore, the basin stability analysis is not only relevant for systems featuring larger number of states, but also for rather low-dimensional systems.

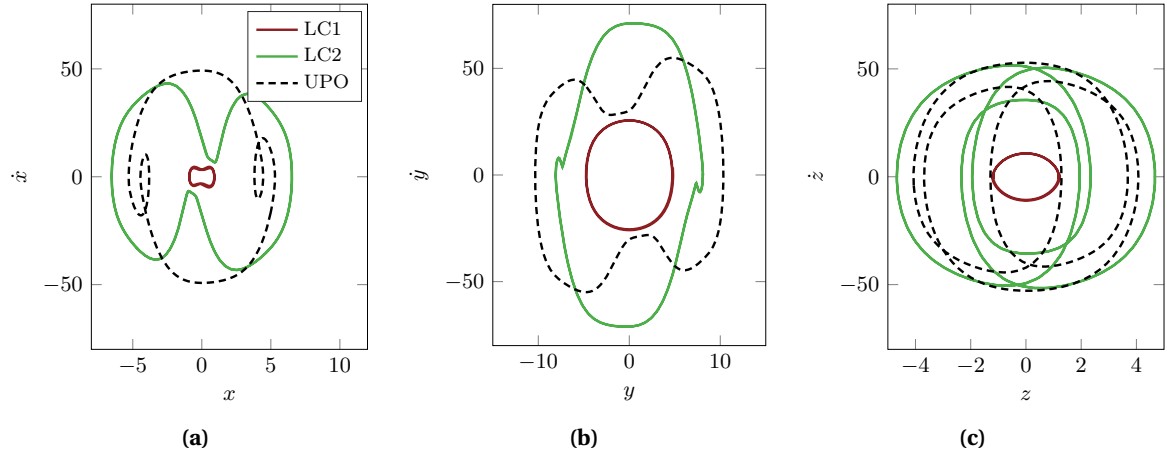

**Figure 8.** State space of all DOFs (horizontal direction in (**a**), vertical direction in (**b**) and diagonal direction in (**c**)) at $k_x = 27.0$ for the weakly damped oscillator.

## 6. Conclusions

This work proposed augmenting the classical local stability analysis of friction-excited oscillators by their basin stability. The concept of basin stability allows assigning global stability metrics to multi-stable solutions in a highly automated manner including error estimates. For three different friction-excited systems, we show that the knowledge of global stability with respect to a specific set of initial conditions can provide important insights into the long-term dynamics. Particularly for well-controlled perturbations, this approach allows estimating the probabilities of arriving on either of multiple stable solutions, and even to rule out some steady-state behavior. As a result, we suggest to include the basin stability analysis into the toolbox of techniques that are applied to study the nonlinear dynamics of multi-stable systems, especially when operating conditions are well-known.

**Author Contributions:** Conceptualization, M.S. and N.H.; methodology, M.S.; software, M.S.; validation, M.S.; formal analysis, M.S.; investigation, M.S. and A.P.; resources, N.H.; data curation, M.S.; writing—original draft preparation, M.S.; writing—review and editing, N.H. and A.P.; visualization, M.S.; supervision, N.H. and A.P.; project administration, N.H.; funding acquisition, N.H. All authors have read and agreed to the published version of the manuscript.

**Acknowledgments:** Publishing fees supported by Funding Programme **Open Access Publishing** of Hamburg University of Technology (TUHH).

**Funding:** This research was funded by the German Research Foundation (Deutsche Forschungsgesellschaft DFG) within Priority Program 1897 'calm, smooth, smart', grant number 314996260.

**Conflicts of Interest:** The authors declare no conflict of interest. The funders had no role in the design of the study; in the collection, analyses, or interpretation of data; in the writing of the manuscript, or in the decision to publish the results.

## Abbreviations

The following abbreviations are used in this manuscript:

DOF    degree-of-freedom
FIV    friction-induced vibrations
FP    fixed point
LC    limit cycle

## Appendix A. Single-DOF Oscillator

*Appendix A.1. Equations of Motion*

Using $\omega_{\mathrm{n}} = \sqrt{\frac{k}{m}}$, $\xi = \frac{c}{2\sqrt{km}}$, $x_0 = \frac{N}{k}$, and $\tau = \omega_{\mathrm{n}} t$, $\frac{\mathrm{d}}{\mathrm{d}t} = \omega_{\mathrm{n}} \frac{\mathrm{d}}{\mathrm{d}\tau}$ we re-write Equation (5) into

$$\ddot{\tilde{x}} + 2\xi \dot{\tilde{x}} + \tilde{x} = \tilde{F} \tag{A1}$$

where $(\tilde{\cdot})$ indicates a non-dimensional quantity.

*Appendix A.2. Convergence of Basin Stability Values*

The number of samples $n$ is varied to answer the question of how many samples from $\mathcal{Q}$ are required for a robust approximation of the basin stability values. Figure A1 displays the convergence of the basin stability values for the single-DOF oscillator case and the corresponding analytical values.

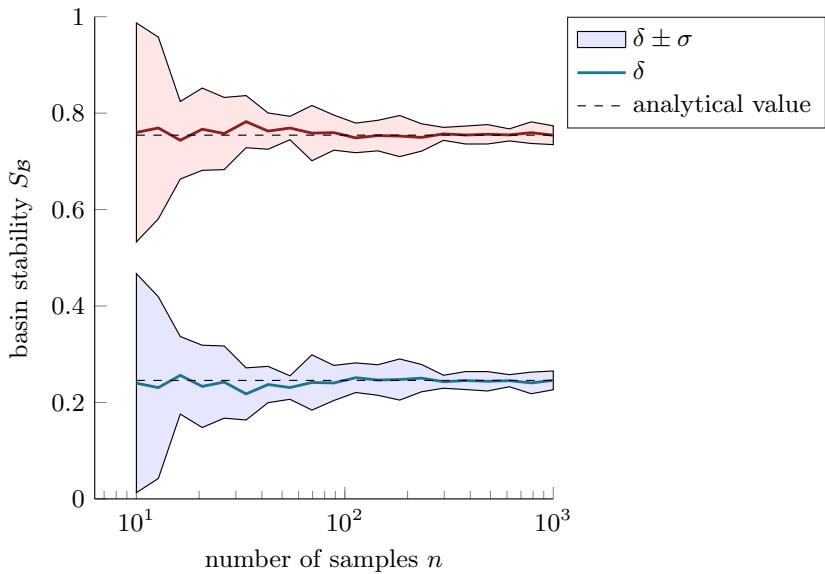

**Figure A1.** Effect of increasing the number of samples for estimating the basin stability values at $\tilde{v}_{\mathrm{d}} = 1.5$. For each value $n$, the calculation has been repeated ten times. Mean values $\delta$ and the standard deviation $\sigma$ are reported along with the analytical values.

## Appendix B. Mode-Coupling Instability Oscillator

The equations of motion are given by

$$\mathbf{M}\ddot{\mathbf{x}} + (\mathbf{D} + \mathbf{G})\,\dot{\mathbf{x}} + (\mathbf{K} + \mathbf{N})\,\mathbf{x} + \mathbf{f}_{\mathrm{nl}} = \mathbf{0}, \quad \mathbf{x} = [x, y, z]^{\top},$$

$$\mathbf{M} = \begin{bmatrix} m & 0 & 0 \\ 0 & m & 0 \\ 0 & 0 & m_1 \end{bmatrix}, \quad \mathbf{D} = \begin{bmatrix} d_{\mathrm{x}} & 0 & 0 \\ 0 & d_{\mathrm{y}} & 0 \\ 0 & 0 & d_{\mathrm{x}} \end{bmatrix}, \quad \mathbf{G} = \mathbf{0}, \quad \mathbf{K} = \begin{bmatrix} k_{\mathrm{x}} & -0.5k_{\mathrm{y}}\mu & 0 \\ -0.5k_{\mathrm{y}}\mu & k_{\mathrm{y}} & 0 \\ 0 & 0 & k_{\mathrm{z}} \end{bmatrix}, \tag{A2}$$

$$\mathbf{N} = \begin{bmatrix} 0 & -0.5k_{\mathrm{y}}\mu & 0 \\ 0.5k_{\mathrm{y}}\mu & 0 & 0 \\ 0 & 0 & 0 \end{bmatrix}, \quad F_{\mathrm{nl}} = uk_{\mathrm{lin}} + u^3 k_{\mathrm{nl}} + \dot{u} d_{\mathrm{lin}}, \quad \mathbf{f}_{\mathrm{nl}} = \begin{bmatrix} -\frac{\sqrt{2}}{2} \\ -\frac{\sqrt{2}}{2} \\ 1 \end{bmatrix} F_{\mathrm{nl}}$$

where $u$ is the relative displacement in the joint between the main mass and the secondary mass, given by $u = -\frac{\sqrt{2}}{2}x - \frac{\sqrt{2}}{2}y + z$. The parameter values for the reference configuration are given by $m = m_1 = 1$, $k_{\mathrm{x}} = 32.5$, $k_{\mathrm{y}} = 20$, $k_{\mathrm{z}} = 100$, $k_{\mathrm{lin}} = 10$, $k_{\mathrm{nl}} = 5$, $d_{\mathrm{x}} = d_{\mathrm{y}} = d_{\mathrm{z}} = d_{\mathrm{lin}} = 0.02$, $\mu = 0.65$.

## Appendix C. Mode-Coupling Instability Oscillator with Isolated Solutions

Compared to the system configuration given in Appendix B, the damping parameter values are set to $d_\mathrm{x} = d_\mathrm{y} = d_\mathrm{z} = d_\mathrm{lin} = 0.002$. Initial values on the periodic orbits at $k_\mathrm{x} = 11$ for the weakly damped system configuration read

$$
\begin{aligned}
\mathbf{y}_0 &= \begin{bmatrix} -1.1366 & -4.5527 & -1.2077 & -0.0125 & 0.0722 & -0.0054 \end{bmatrix}^\top \quad \text{LC1} \\
\mathbf{y}_0 &= \begin{bmatrix} 5.9650 & -6.6938 & -4.5901 & 0.2163 & 8.6960 & -6.1122 \end{bmatrix}^\top \quad \text{LC2} \,.
\end{aligned}
\tag{A3}
$$

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
