# Peer review of "The Basin Stability of Bi-Stable Friction-Excited Oscillators"

_lubricants, doi:10.3390/lubricants8120105_

Round 1

Reviewer 1 Report

The authors studied the basin stability of bi-stable friction-excited oscillators. The authors' work is significant, and the study work is very complete. I think that it can be accepted for publication. The authors mostly analyzed a single degree of freedom in the paper. It is easy to be extended to the stability analysis of multiple degrees of freedom? It is can be also used to identify which one of several stability vibration modes is mostly likely to occur? Please response.

Author Response

We thank the reviewer for this feedback.

Question 1 (multiple DOFS): In fact, one of the major advantages of the Monte Carlo simulation-based basin stability approach is the applicability to systems of multiple and many degrees of freedom. Except for the accordingly scaling number of computations required, there is no restriction to the size of the dynamical system. We added the following sentence to the Methods section: “The size, i.e. the number of states, of the dynamical system to be studied by the basin stability is only limited by computational power for the Monte Carlo simulations.”

Question 2 (solution probability): The concept of basin stability can be understood as a direct way for inferring the most likely solution in a multi-stable solution (e.g. see Conclusion: “… this approach allows to estimate the probabilities of arriving on either of multiple stable solutions, and even to rule out some steady-state behavior.”). If the reviewer is referring to stable fixed point solutions that may become unstable (resulting in vibrations) under some parametric change, local stability metrics such as the instantaneous Lyapunov exponent would be a better candidate. The basin stability cannot yield information about the distance to the stability boundary for some solution

Reviewer 2 Report

The manuscript under review represents a completed study which is of scientific and practical significance. It can be recommended for publication. The only comment is that the model of Eq.(5) has been known as minimum as from the middle of the 20th century. In particular, it was reported in

I.V. Kragelskii, Friction and wear, 2d ed., Mashinostroenie, Moscow, 1968. (in Russian)

Translation into English:

I.V. Kragelskii, Friction and wear, Butterworth, London 1965.

In comparatively recent literature, please, see

K. Nakano, S. Maegawa, Safety-design criteria of sliding systems for preventing friction-induced vibration, Journal of Sound and Vibration 324 (2009) 539–555.

Best regards!

Author Response

We thank the reviewer for pointing us to these pieces of relevant and related work. We have included the second reference in the revised manuscript.